

# Kill two birds with one stone: making multi-transgenic pre-diabetes mouse models through insulin resistance and pancreatic apoptosis pathogenesis

Siyuan Kong[1,2,*], Jinxue Ruan[1,*], Kaiyi Zhang[1], Bingjun Hu[1], Yuzhu Cheng[1], Yubo Zhang[2], Shulin Yang[1] and Kui Li[1,2]

[1] State Key Laboratory of Animal Nutrition & Key Laboratory of Farm Animal Genetic Resource and Germplasm Innovation of Ministry of Agriculture, Institute of Animal Sciences, Chinese Academy of Agricultural Sciences, Beijing, Beijing, China
[2] Agricultural Genomics Institute at Shenzhen, Chinese Academy of Agricultural Sciences, Shenzhen, Guangdong, China
[*] These authors contributed equally to this work.

Corresponding author
Shulin Yang, yangshulin@caas.cn

## ABSTRACT

**Background**. Type 2 diabetes is characterized by insulin resistance accompanied by defective insulin secretion. Transgenic mouse models play an important role in medical research. However, single transgenic mouse models may not mimic the complex phenotypes of most cases of type 2 diabetes.

**Methods**. Focusing on genes related to pancreatic islet damage, peripheral insulin resistance and related environmental inducing factors, we generated single-transgenic (C/EBP homology protein, CHOP) mice (CHOP mice), dual-transgenic (human islet amyloid polypeptide, hIAPP; CHOP) mice (hIAPP-CHOP mice) and triple-transgenic (11β-hydroxysteroid dehydrogenase type 1, 11β-HSD1; hIAPP; CHOP) mice (11β-HSD1-hIAPP- CHOP mice). The latter two types of transgenic (Tg) animals were induced with high-fat high-sucrose diets (HFHSD). We analyzed the diabetes-related symptoms and histology features of the transgenic animals.

**Results**. Comparing symptoms on the spot-checked points, we determined that the triple-transgene mice were more suitable for systematic study. The results of intraperitoneal glucose tolerance tests (IPGTT) of triple-transgene animals began to change 60 days after induction ($p < 0.001$). After 190 days of induction, the body weights ($p < 0.01$) and plasma glucose of the animals in Tg were higher than those of the animals in Negative Control (Nc). After sacrificed, large amounts of lipid were found deposited in adipose ($p < 0.01$) and ectopically deposited in the non-adipose tissues ($p < 0.05$ or 0.01) of the animals in the Tg HFHSD group. The weights of kidneys and hearts of Tg animals were significantly increased ($p < 0.01$). Serum C peptide (C-P) was decreased due to Tg effects, and insulin levels were increased due to the effects of the HFHSD in the Tg HFHSD group, indicating that damaged insulin secretion and insulin resistance hyperinsulinemia existed simultaneously in these animals. The serum corticosterone of Tg was slightly higher than those of Nc due to the effects of the 11βHSD-1 transgene and obesity. In Tg HFHSD, hepatic adipose deposition was more severe and the pancreatic islet area was enlarged under compensation, accompanying apoptosis. In the transgenic control diet (Tg ControlD) group, hepatic adipose deposition was also severe, pancreatic islets were damaged, and their areas were

decreased ($p < 0.05$), and apoptosis of pancreatic cells occurred. Taken together, these data show the transgenes led to early-stage pathological changes characteristic of type 2 diabetes in the triple-transgene HFHSD group. The disease of triple-transgenic mice was more severe than that of dual or single-transgenic mice.

**Conclusion**. The use of multi-transgenes involved in insulin resistance and pancreatic apoptosis is a better way to generate polygene-related early-stage diabetes models.

## INTRODUCTION

Type 2 diabetes mellitus (T2DM), which is characterized by peripheral insulin resistance and impaired insulin secretion, is a chronic metabolic disease that has shown an increased incidence in obese and aged individuals in recent years (*American Diabetes Association, 2009*; *Drouin et al., 2009*). T2DM is a multifactorial disease that is associated with genetic factors such as susceptibility genes and environmental factors such as intake of high-fat and high-sucrose diets (*Nath, Ghosh & Choudhury, 2016*; *Qiu, Moore & Darabos, 2016*). β-cell failure and peripheral insulin resistance are pathogenic features of T2DM (*Kahn, 2003*; *Lee & Cox, 2011*). Several mechanisms may be responsible for the progressive β-cell failure and insulin resistance that occurs in T2DM, including long-term β-cell stress, apoptosis, functional exhaustion and enhanced glucocorticoid levels (*Höppener, Ahrén & Lips, 2000*; *Höppener & Lips, 2006*; *Zhang et al., 2016*). It was reported that β-cell endoplasmic reticulum stress (ERS) can induce cell apoptosis and insulin secretion deficiency in pancreatic cells (*Zhang et al., 2016*). C/EBP homology protein (CHOP) was discovered as a direct upstream factor that drives ERS and apoptosis (*Oyadomari & Mori, 2004*). In addition, islet amyloid polypeptide (IAPP) levels have been commonly demonstrated in patients who are obese or who display damaged glucose secretion or glucose intolerance (*Hartter et al., 1991*). IAPP, also referred as amylin, is the primary component of the insoluble pancreatic amyloid fibrils in T2DM (*Clark et al., 1987*). The human gene hIAPP encodes amylin, which accumulates in patients' pancreatic β-cells (*Costes et al., 2013*; *Hull et al., 2013*). It is known that this phenomenon can induce the β-cell unfolded protein response (UPR) and apoptosis (*Khemtémourian et al., 2008*; *Meier et al., 2007*). Moreover, 11β-hydroxysteroid dehydrogenase type 1 (11β-HSD1) is an important dehydrogenase that is associated with insulin resistance (*Peng et al., 2016*; *Pereira et al., 2012*). 11β-HSD1 overexpression can increase glucocorticoid levels (*Pereira et al., 2012*) in liver. Insulin receptor desensitization and glucose absorption reduction are promoted by glucocorticoids, which can also inhibit β-cell insulin secretion and disrupt normal insulin-plasma glucose balance (*Johnson et al., 1992*; *Lee & Cox, 2011*; *Masuzaki & Flier, 2004*).

Transgenic animal models are needed in T2DM research. As a disease to which susceptibility is controlled by multiple genes (polygenes), T2DM is to a certain degree
hereditary (*Qiu, Moore & Darabos, 2016*). Most currently available mouse diabetes models are single-gene engineered. Several mouse models of insulin resistance have been described (*Nandi et al., 2004*). Pioneristic studies aimed at understanding the pathogenesis of insulin resistance targeted the insulin receptor ubiquitously or in specific tissues. Moreover, further studies addressed many components of the insulin signaling pathway, as well as nuclear factors involved in both insulin action and beta cell function, such as HMGA1 (*Arcidiacono et al., 2015*; *Foti et al., 2005*), or the transcription factor FoxO1 (*Nakae et al., 2002*). Insulin signal transduction defects will lead to insulin resistance. The insulin receptor (IR) knock-out mice are basing on this mechanism. They were the first genetically engineered mouse models for T2DM. For IR +/- deficient mice, initially, there was no obvious metabolic abnormality. Only 10% of them developed to diabetes in adulthood (*Brüning et al., 1997*). For IR -/- deficient mice, during lactation, they suffered from abnormal metabolism, delayed growth and skeletal muscle atrophy, accompanying elevated plasma triglycerides and free fatty acids. Eventually, they died within 1 week after birth, resulting from diabetic ketosis acidosis (*Accili et al., 1996*). Moreover, leptin-deficient ob/ob mice and leptin receptor-deficient db/db mice were important for their wide use in obesity-induced T2DM (*Drel et al., 2006*), although leptin or leptin receptor disorder is not a key point in human T2DM (*Wang, Chandrasekera & Pippin, 2014*). For reduced pancreatic β-cell secretion, Johnson et al. generated Pdx-1 +/- mice, which exhibited increased islet apoptosis, reduced insulin release but normal β-cell function (mimicking maturity onset diabetes in the young, MODY-4) (*Masiello, 2006*). *Fajans, Bell & Polonsky (2001)* prepared the hepatocyte nuclear factor-1 (HNF-1) transgenic mice similar to MODY-3, which had defective insulin secretory responding to glucose (*Masiello, 2006*). However, HFN-1 mainly worked to regulate the differentiation rather than the mass of β-cells (*Masiello, 2006*). Because it is difficult for these mouse models to comprehensively describe the characteristics of a disease to which susceptibility is polygenic, methods for obtaining more appropriate genetically modified disease models are important research priorities. During our work in transgenic disease model construction, we conceived the idea of combining the functional genes involved in insulin secretion defects and peripheral insulin resistance together at the genetic level (*Kong et al., 2016*; *Kong et al., 2015*; *Lee & Cox, 2011*). Although there is no direct evidence in the literature concerning the interaction of the three (11βHSD1, hIAPP and CHOP) aforementioned genes in diabetes pathogenesis, they are all important genes that can lead to diabetes (*Matveyenko & Butler, 2006*; *Oyadomari & Mori, 2004*; *Pereira et al., 2012*). Thus, mouse models of the disease phenotype might be developed by combing these three genes. In our groundbreaking research, the use of multi-transgenes may offer a better way to "kill two birds with one stone" for generating polygene-related diabetes models.

## METHODS

### Transgenic mice

Three kinds of single, dual and triple transgenic C57BL/6 mice with a porcine apolipoprotein E promoter fragment linked to the 11β-HSD1 gene and/or a porcine insulin

promoter fragment linked to the CHOP gene and the hIAPP gene were generated. pGL3-PIP-CHOP is a single transgene vector in which the CHOP gene is driven by pancreas-specific PIP (porcine insulin promoter). pGL3-PIP-hIAPP-F2A-CHOP is a dual-gene polycistronic system in which the two genes hIAPP and CHOP are connected to *Furin*-2A and are driven together by the PIP. pcDNA3.1-PapoE-11βHSD1-PIP-CHOP-F2A-hIAPP is a tissue-specific polycistronic system in which 11β-HSD1 is driven by the liver-specific PapoE (porcine apoE promoter) (*Xia et al., 2014*) and hIAPP and CHOP are linked to the F-2A peptide, which is driven by the pancreas-specific PIP (*Kong et al., 2016*). The PapoE (porcine apolipoprotein E promoter) sequence (*Xia et al., 2014*), the PIP (porcine insulin promoter) sequence (*Kong et al., 2016*), the 11β-HSD1 gene sequence (GenBank: NM_214248.1), CHOP gene sequence (GenBank: NM_007837.3) and the hIAPP gene sequence (GenBank: NM_000415.2) are stored in the National Center for Biotechnology Information (NCBI) database. The vectors pGL3-PIP-CHOP, pGL3-PIP-hIAPP-F2A-CHOP and pcDNA3.1-PapoE-11βHSD1-PIP-CHOP-F2A-hIAPP were synthesized by Generay Biotech Co. Ltd. (Shanghai, China). The linear DNA sequence was microinjected into the pronuclei of zygotes of C57BL/6 mice. Positive males from the F0 generation of transgenic mice were bred to wild-type female mice purchased from Vital River Laboratory Technology Co. Ltd. (Beijing, China). Finally, positive male F1 generation mice were obtained. The related positive identification primers are list in Table S1. The transgenic and control mice used in the four treatments were male. The mice were allowed free access to food and water and were maintained at a temperature of 20–22 °C, relative humidity of 30–70%, and a 12-h light/dark cycle. The animals received humane care according to the recommendations in the Guide for the Care and Use of Laboratory Animals. All procedures were approved by the Animal Care and Use Committee of the Germplasm Resource Center (Institute of Animal Sciences, Chinese Academy of Agricultural Sciences, Beijing, China) (permit no. ACGRCM 2013-035).

## High-fat high-sucrose diet (HFHSD) induction strategy

In the research, single-transgenic mice (CHOP), dual-transgenic mice (hIAPP-CHOP) and triple-transgenic mice (11β-HSD1-hIAPP-CHOP) were generated. Single-transgenic mice (CHOP) were fed a control diets (Research Diets, D12450K) ad libitum. The dual-transgenic and triple-transgenic mice were fed high-fat high-sucrose diets from the age of 13 weeks until sacrifice. At 12 weeks of age, the positive (Tg) mice and the control (Nc) mice were divided into two groups ($n = 5$–6) for acclimatization. At 13 weeks, the animals were subdivided into four groups as follows: Tg animals fed a high-fat high-sucrose diet (Research Diets, D12451) (Tg HFHSD); a negative control group fed a high-fat high-sucrose diet (Nc HFHSD); a Tg group fed a control diet (Research Diets, D12450K) (Tg ControlD); and a negative control group fed a control diet (Nc ControlD). The diets were purchased from Research Diets, Inc., USA. The induction time was 190 days.

## Glucose, body weight and IPGTT

Body weights were monitored daily for 30 days. Prior to the IPGTT experiment, the mice were weighed using an electronic balance. The animals' fasting plasma glucose

concentrations were determined by random inspection using the glucose oxidase method (One Touch® Ultra; One Touch, Tampa, FL, USA). IPGTT was performed at the beginning of diet induction (at 13 weeks of age, induced 0 days) and again at 60 days and 190 days of induction. The mice were fasted for 12-14 h prior to IPGTT testing (drinking water was maintained, and food was removed). A volume of 20% glucose solution equal to 1% of the weight of the animal was injected into the intraperitoneal (for example, 50 g x 1%, inject 0.5 ml). Then, the glucose concentrations in blood taken from the tail vein were measured using a One Touch glucometer at time points of 0 min, 15 min, 30 min, 45 min, 60 min, 90 min and 120 min.

### Anatomy and sampling

Mice were sacrificed by cervical dislocation after plasma sampling and photographed immediately in front and back views. The pancreas, liver, other viscera and adipose tissue were collected and divided into two parts. Tissues for hematoxylin-eosin staining (HE) and immunohistochemistry were fixed in 4% paraformaldehyde (PFA), and the second portion was immediately frozen and stored at −80 °C.

### Serological measurements

Blood samples were taken from the retrobulbar intraorbital vessels of the animals before sacrifice. The samples were placed in 1.5-ml sterile eppendorf tube (EP) tubes (without heparin sodium) and stored on ice. Serum was obtained by centrifugation (12,000 rpm, 4 °C). serum glucose (GLU), high-density lipoprotein cholesterol (HDL-C), low-density lipoprotein cholesterol (LDL-C), corticosterone (COR), triglycerides (TG) and C peptide (C-P) were determined using a HITACHI 7080 automatic biochemical analyzer (Hitachi, Ltd.; One Touch, Japan). Insulin (INS) was measured using a DFM-96 radioimmunoassay gamma counter (Hitachi, Ltd. Japan).

### HE and immunohistochemistry

To further determine the pathological changes in the livers (left lobe) and islets (pancreatic tail), the animals' livers and pancreases were embedded in paraffin and sectioned at 5-8 μm. The samples were subjected to conventional hematoxylin and eosin staining using routine methods (*Ruan et al., 2016*). The numbers of liver lipid vacuoles in each group were counted, and their corresponding area ratios were calculated and binned based on size from "<1,500" to ">70,000" in a series of 10 intervals. The immunohistochemistry was tested for islets, and insulin secretion was assessed using an antibody to pancreatic insulin (ab63820; Abcam, Cambridge, UK). The AOD of pancreatic insulin expression was calculated as the IOD (integral optical density) sum/Area sum (*Hu, 2014*; *Ruan et al., 2016*). To detect amylin deposition within islets in the pancreas, immunohistochemical analysis of IAPP was performed using an anti-amylin antibody (ab115766; Abcam, Cambridge, UK). Apoptotic cells in the pancreas were detected using an antibody against caspase 3 (ab217550; Abcam, Cambridge, UK). An Olympus microscope (CX31; Olympus Corporation, Tokyo, Japan) with a Pixera digital camera (Pro 120es; Pixera Corporation, San Jose, CA, USA) was used to photograph the sections. The procedures have been described in detail previously (*Kong et al., 2016*).

## Statistical analysis

The data were analyzed using SPSS v.22.0 (IBM, USA). The figures were drawn using Graph Pad Prism 5 (GraphPad Software, Inc., USA). The data are presented as the mean ± SEM. Comparison of the differences between two groups was made using an unpaired, one-tailed Student's $t$ test. One-way ANOVA analysis of variance and Tukey's test were performed for visceral organ and adipose tissue weight comparisons. $P$ values of <0.05 were regarded as statistically significant (* $p < 0.05$, ** $p < 0.01$, *** $p < 0.001$). The average optical density values (AOD): "IOD sum" and "Area sum" were calculated using Image-Pro Plus 6.0 software (Media Cybernetics, Inc, Rockville, MD, USA).

## RESULTS

### Generation and identification of single-transgenic, dual-transgenic and triple-transgenic mice

Because diabetes is a disease involving many metabolic pathways, prominent among which are damaged central insulin secretion and accompanying peripheral insulin resistance (*Basevi, 2012*), we constructed diabetes-related gene engineering mouse models containing either one transgene (CHOP), two transgenes (hIAPP and CHOP) or three transgenes (11β-HSD1,CHOP and hIAPP). To do this, the vectors pGL3-PIP-CHOP, pGL3-PIP-hIAPP-F2A-CHOP and pcDNA3.1-PapoE-11βHSD-PIP-CHOP-F2A-hIAPP were constructed (Fig. 1A). Gene expression from the first and second vector sequences is controlled by the pancreas-specific insulin promoter. It was supposed that expression of the transgene specifically promotes pancreatic islet β cell apoptosis, leading to damaged insulin secretion. The third vector contains a liver-specific apoE promoter and a pancreas-specific insulin promoter; together, these promoters promote hepatic insulin resistance and damage pancreas insulin secretion simultaneously. Agarose gel electrophoresis of the genomic PCR products showed positive identification results (Figs. 1B–1D). The primers and methods have been described previously (*Kong et al., 2016*).

### Comparison of plasma glucose related symptom at the spot-checking time in single-transgenic, dual-transgenic and triple-transgenic mice

Single-transgenic CHOP mice were observed until five months of age without any induction. At that time, intraperitoneal glucose tolerance tests (IPGTT) were administered (Fig. 2A, Single-Tg). Administration of the IPGTT rapidly increases plasma glucose levels, which should stimulate the secretion of insulin, which in turn reduces plasma glucose levels. If the plasma glucose was still very high after the test or returned only slowly to normal levels, the animal was considered insulin resistant. The plasma glucose change showed mild GTT damage in the Single-Tg group, but these animals recovered normal levels of plasma glucose (7.17 ± 0.35 mmol/l) at 120 min. Compared with 5-month-old triple-transgenic mice (Fig. 2A, Triple-Tg), whose IPGTT 120-minute glucose level was 9.72 ± 0.77 mmol/l, this showed that the effect of a single transgene was not as great as the effect of the triple transgene ($p < 0.05$ vs Single-Tg). The corresponding area under the curve was calculated; the AUC of Triple-Tg was larger than that of Single-Tg and significantly larger than that of Nc ($p < 0.05$ vs Nc group) (Fig. 2B, Triple-Tg). Furthermore, after HFHSD induction

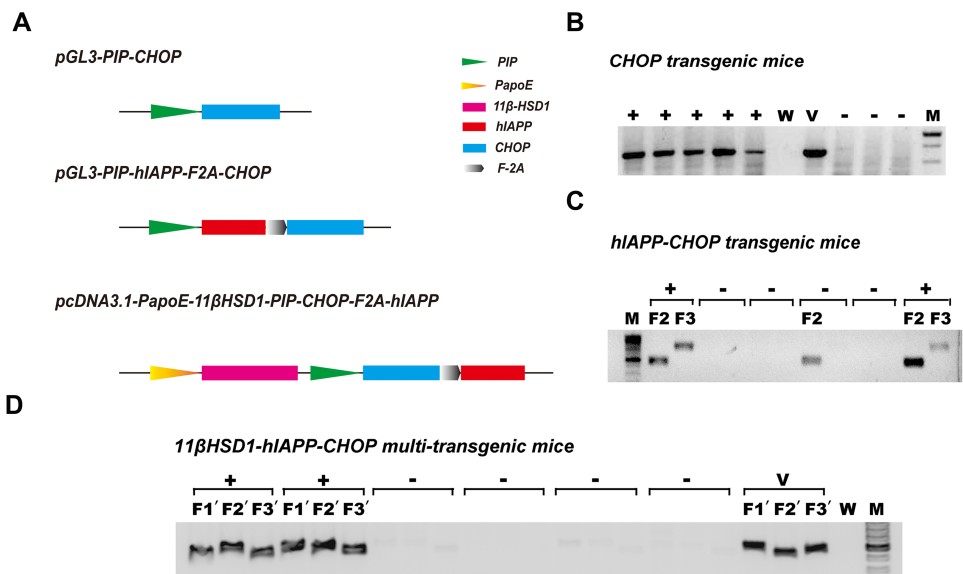

**Figure 1 Schematic diagram of the vectors used in genetic engineering of mice and identification of positive transgenic animals.** (A) Schematic structures of the vectors containing the three genes (CHOP, hIAPP-CHOP and 11βHSD1-hIAPP-CHOP) used to create transgenic mice. (B) Genomic PCR positive amplification of the CHOP transgene. +, positive; −, negative; W, ddH$_2$O; V, positive plasmid vector; M, 100-bp DNA marker. (C) Genomic PCR positive amplification of animals carrying the CHOP (F2) and hIAPP (F3) transgenes. Mice were regarded as positive (+) when the two bands (F2 and F3) were present at the same time; otherwise, they were considered negative (−). (D) Genomic PCR positive amplification of 11β-HSD1 (F1′), CHOP(F2′) and hIAPP(F3′). If the three bands F1′, F2′ and F3′ of the object were present at the same time, the animal was considered positive. V, positive plasmid vector; W, ddH$_2$O; M, 100-bp DNA marker.

for approximately 190 days, the 120-minute IPGTT glucose level of triple-transgenic mice ($16.3 \pm 0.76$ mmol/l) was higher than that of dual-transgenic mice of the same age, ∼11 months, which was $12.47 \pm 1.61$ mmol/l, lower than 15 mmol/l (Fig. 2C, Triple-Tg HFHSD $p < 0.01$ vs Nc HFHSD). The corresponding areas under the curve were also calculated. The AUC of Triple-Tg HFHSD was larger than those of Double-Tg and Nc HFHSD ($p < 0.05$ vs Nc HFHSD) (Fig. 2D, Triple-Tg).

In humans, the current diagnostic criteria for fasting plasma glucose (FPG) are 7.0 mmol/l for FPG and 11.1 mmol/l for the 120-min PG value (*Basevi, 2012*). We compared the FPG of three kinds of transgenic mice at the time of sacrifice (Fig. 2E). The glucose levels of triple-transgenic and dual-transgenic mice were indeed increased above 7.0 mmol/l by ∼2 mmol/l. Consequently, in further experiments, we focused on the triple-transgenic animals.

## Analysis of early-stage diabetes-related phenotypes in triple-transgenic mice fed a high-fat, high-sucrose diet
### Disease symptoms related to early-stage T2DM

The triple-transgenic mice were fed control diets ad libitum until 13 weeks of age. At 13 weeks of age, some of the animals were placed on HFHS diets to enhance the food and

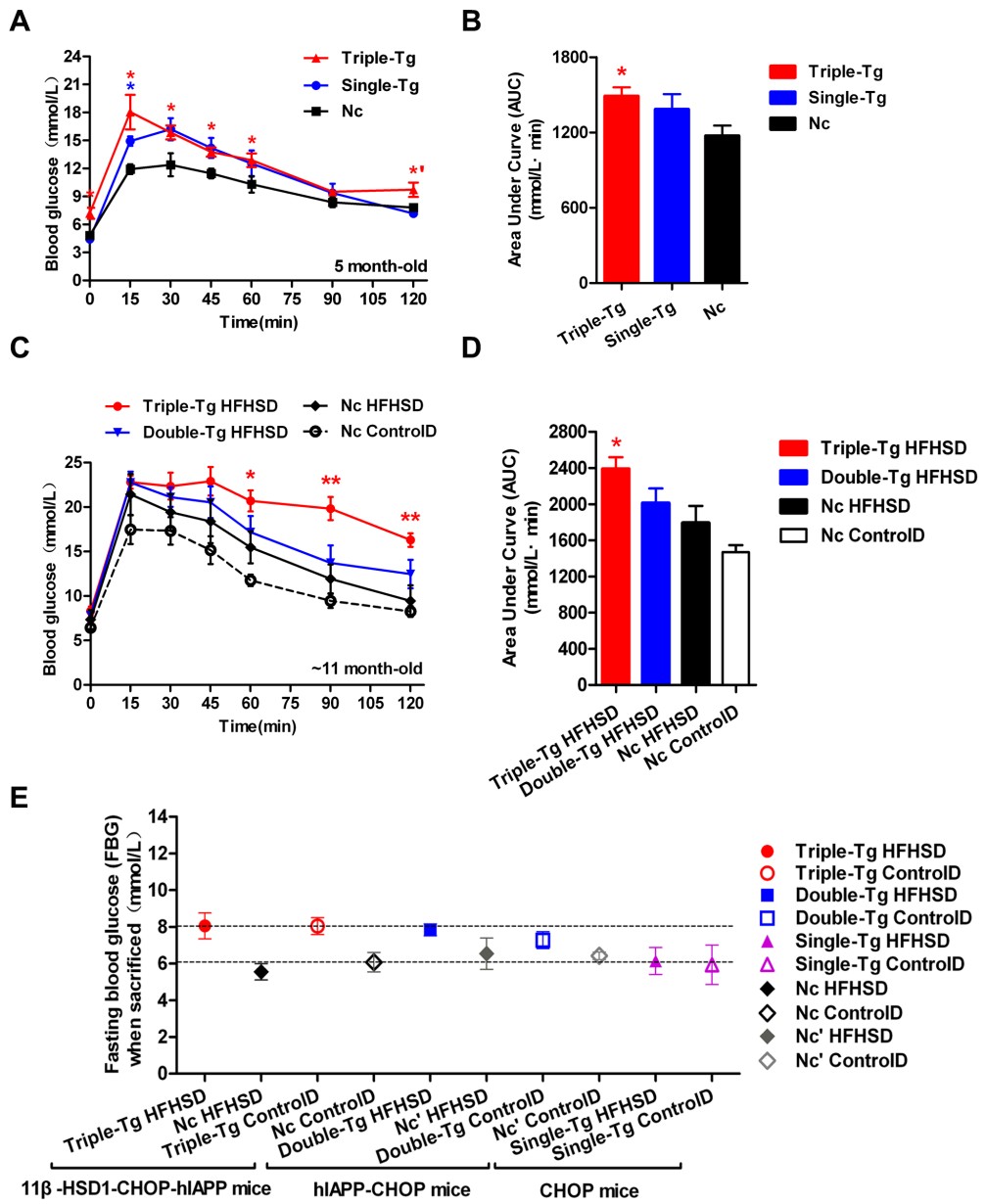

**Figure 2  Comparison of plasma glucose levels in single-transgenic, dual-transgenic and triple-transgenic mice.** (A) IPGTT without HFHSD diet induction (5-month-old animals). Single-Tg (blue), CHOP-transgenic mice; Triple-Tg (red), 11β-HSD1-CHOP-hIAPP transgenic mice; Nc (dark), negative control. $n = 3$–5. * $p < 0.05$ vs Nc group. *' $p < 0.05$ vs Single-Tg. (B) Area under the curve (AUC) of IPGTT of (A). (C) IPGTT with HFHS diet induction (11-month-old animals). Triple-Tg HFHSD (red), 11β-HSD1-CHOP-hIAPP transgenic mice with HFHS diet induction; Double-Tg HFHSD (blue), hIAPP-CHOP transgenic mice with HFHS diet induction; Nc HFHSD (dark), negative control mice with HFHS diets induction; Nc ControlD (white), negative control mice with control diets. * $p < 0.05$ or ** $p < 0.01$ vs Nc HFHSD Group; $n = 4$–13. (D) Area under the curve (AUC) of IPGTT of (C). (E) Fasting plasma glucose levels of the three kinds of transgenic mice at the time of sacrifice. The gray symbols indicate Nc mice (HFHS diet-induced or not induced). The colored symbols indicate Tg mice (HFHS diet-induced or not induced). $n = 5$–6. The data are shown as the mean ±SEM. $P$ values were calculated using Student's $t$-test.

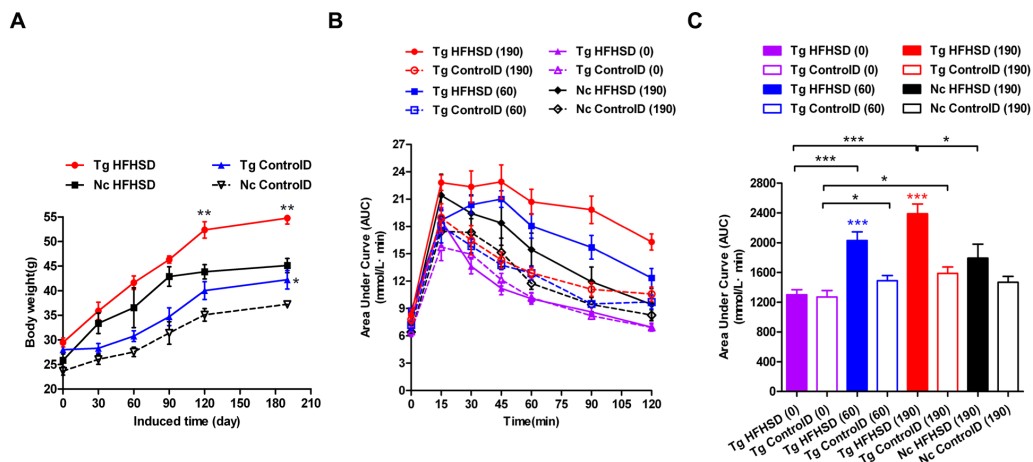

**Figure 3** **Body weights and IPGTT data indicating insulin resistance of triple-transgenic mice.** (A) The increasing weight trends of triple-transgenic mice during the induction period (the induction period lasted for 190 days and was started when the mice were 13 weeks of age. Tg HFHSD (red), 11β-HSD1-CHOP-hIAPP triple-transgenic mice with HFHS diet induction; Tg ControlD (blue), triple-transgenic mice fed a control diet; Nc HFHSD (black), negative control mice with HFHS diet induction; Nc Control D (white), negative control mice fed a control diet. ** $p < 0.01$ vs Nc HFHSD group; * $p < 0.05$ vs Nc ControlD group; $n = 3–6$. (B) IPGTT dynamic trends of triple-transgenic mice under HFHS diet induction. The checkpoints include induction for 0 days, 60 days and 190 days. The results for the Nc HFHSD group and the Nc ControlD group at 190 days of induction are included as negative references. $n = 4–6$ (C) The area under the curve (AUC) of (B). The data are presented as the mean ± SEM. Significance levels are indicated by * $p < 0.05$, ** $p < 0.01$ and *** $p < 0.001$. $P$ values were calculated using Student's $t$-test.

drink effect. The food and drink effect led to obesity, which is one of the most important environmental factors associated with type 2 diabetes. Obese individuals with genetic susceptibility to type 2 diabetes very often develop this disease (*Basevi, 2012*). It has been reported that obesity causes some degree of insulin resistance, and a large number of patients with type 2 diabetes are obese (*Basevi, 2012*). Weight tracking showed that the animals in the Tg HFHSD group gained more weight than the control animals from the beginning (starting point: 0 days induction) to the end of the induction period (terminal point: 190 days) (Fig. 3A, $p < 0.01$ vs Nc HFHSD). A significant difference ($p < 0.01$) in body weight was found at 120 days and at 190 days (the time of sacrifice); the weight of the mice in the Tg HFHSD group was larger than that of the mice in the Nc HFHSD group. At the time of sacrifice, the average weight of the mice in the Tg HFHSD group was nearly 55 g more than that of the mice that were fed a normal control diet. Whether induced or not, the weight of the Tg mice (Tg HFHSD group and Tg ControlD group) was larger than that of the Nc mice ($p < 0.01$ or $p < 0.05$) (Fig. 3A).

IPGTT was performed to assess insulin resistance (Figs. 3B and 3C) in the Triple-Tg mice. The area under the IPGTT curve (AUC) for animals that received each of the four treatments was measured at 0, 60 and 190 days (Fig. 3C). At the beginning (0 days), the plasma glucose curve obtained from IPGTT showed no abnormalities in glucose levels in the Tg group. Plasma glucose recovered to normal levels (5–6 mmol/l) at 120 min (Fig. 3C, $AUC_{Tg\ HFHSD(0)}$ vs $AUC_{Tg\ ControlD(0)}$, $p > 0.05$). After 60 days of HFHSD induction, the

animals that received HFHSD showed damaged glucose tolerance, especially those in the Tg HFHSD group (Fig. 3C, $AUC_{Tg\ HFHSD(60)}$ vs $AUC_{Tg\ HFHSD(0)}$, $p < 0.001$); the peak glucose appeared retarded at 45 min (Fig. 3B). These results indicate that the Tg mice that were subjected to HFHSD challenge showed impaired glucose tolerance (Fig. 3C $AUC_{TgHFHSD(60)}$ vs $AUC_{Tg\ ControlD(60)}$, $p < 0.001$). At 190 days of induction, the Tg HFHSD group showed more serious damage (Fig. 3B); the IPGTT 120-minute glucose level was 16.3 ± 0.76 mmol/l. For Tg animals that were induced ($p < 0.001$, vs $AUC_{Tg\ HFHSD(0)}$) and not induced ($p < 0.05$, $AUC_{Tg\ ControlD(0)}$), the AUC was found to increase as a function of the induction time. At 190 days of induction (∼11-month-old animals), the AUC significantly increased ($p < 0.001$, vs $AUC_{Tg\ HFHSD(0)}$). At this time, the AUC of the Tg HFHSD group was significantly higher than that of the Nc HFHSD group ($p < 0.05$).

*Morphological and anatomical analysis*

Patients who are susceptible to type 2 diabetes may have an increased percentage of body fat, predominantly visceral fat (*Basevi, 2012*). Triple-transgenic mice were dissected at 190 days of induction. The animals were photographed prior to sacrifice to demonstrate their morphology (Fig. 4A). The morphology of the animals was consistent with their body weights (Fig. 4B). The mice in the Tg HFHSD group were significantly larger than those in the Nc HFHSD group ($P < 0.01$) and very significantly larger than those in the Nc ControlD group ($P < 0.001$). The visceral organs and adipose tissues of the four groups of animals were dissected (Fig. 4C). Excessive fat deposits, especially of abdominal subcutaneous fat ($p < 0.01$), abdominal visceral fat, perirenal fat ($p < 0.01$), mesenteric adipose tissue ($p < 0.01$) and pericardial adipose tissue ($p < 0.05$), were found in the Tg HFHS group. Typical photographs of these animals are shown in Fig. 4C. The kidneys and hearts of the animals in the Tg HFHS group also weighed significantly more ($p < 0.01$) than those of the animals in the other groups, similar to findings reported in previous research using a miniature pig model of early-stage diabetes (*Li et al., 2015*; *Xia et al., 2015*).

*Serological analysis*

The serum-related parameters associated with diabetes phenotype, insulin resistance and insulin secretion of triple-transgenic mice were measured (Table 1). Under the challenge of HFHSD treatment, the GLU of the Tg group was slightly higher than that of the Nc group. The C-P level was decreased compared with that of the Nc HFHSD group, indicating dysfunctional insulin secretion in the Tg animals (*Hope et al., 2016*). Although HFHSD can increase insulin secretion (the C-P level of the Nc HFHSD group increased), the Tg effect damages it (the Tg HFHSD and Tg ControlD groups' C-P values were lower). However, Tg HFHS INS was increased, indicating the presence of hyperinsulinemia associated with insulin resistance and its cumulative effect. The COR was slightly higher in the Tg HFHS group than in the Nc group due to the combined effects of the transgenes and obesity (COR in the Tg ControlD group was also higher, mostly due to the effects of the 11β-HSD1 transgene). HDL-C was low in the Tg ControlD group ($p < 0.05$), and there were reduced trends of "good cholesterol" in both Tg groups resulting from the Tg effect. The TG level in the Tg animals was slightly increased, indicating increased serum lipid content. This was probably due to the increased liver lipid deposition (TG) mass, which

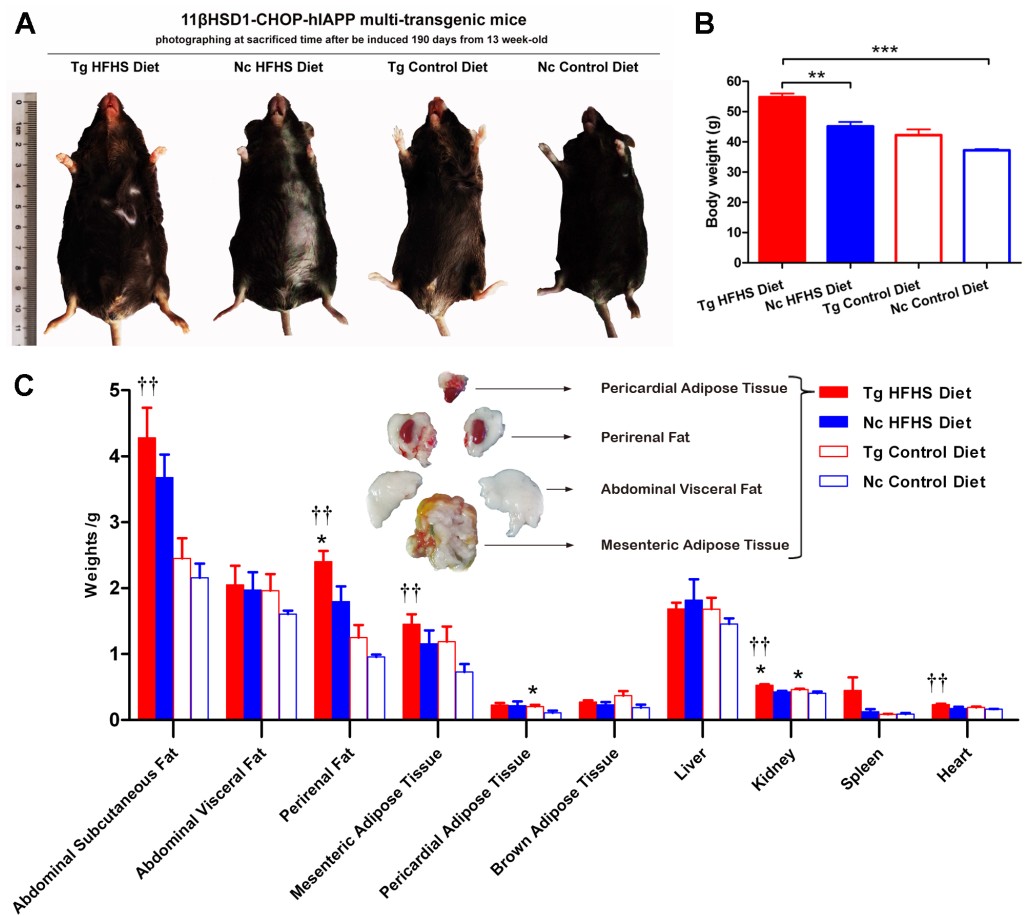

**Figure 4** **Anatomical analysis of triple-transgenic mice.** (A) Photographs of triple-transgenic mice at the time of sacrifice. Representative mice from the four treatment groups are shown together with a scale (∼11 cm). Tg, $n = 5$; Nc, $n = 3$–6. The photo of the mouse was captured by first author Siyuan Kong. (B) Body weight comparison of triple-transgenic mice at the time of sacrifice. The significance levels are indicated by ** $p < 0.01$ and *** $p < 0.001$. $P$ values were calculated using the Turkey's multiple comparison test. (C) Visceral organ and adipose tissue weight comparison. The histogram lists adipose tissues (abdominal subcutaneous fat, abdominal visceral fat, perirenal fat, mesenteric adipose tissue, pericardial adipose tissue and brown adipose tissue) and visceral organs (liver, kidney, spleen and heart) of the four treatments. Some of the adipose tissues obtained from the prominent Tg HFHSD group are shown in the inset. Tg: $n = 5$; Nc: $n = 3$–6. Significant differences between the transgenic group (Tg) and the control group (Nc) are indicated by * $p < 0.05$, ** $p < 0.01$. Significant differences between the Tg HFHSD group and the Nc Control Diet group are indicated by †$p < 0.05$, ††$p < 0.01$. $P$ values were calculated using Student's $t$-test. The data are expressed as the means ± SEM.

was transported through the plasma circulation in Tg individuals (11β-HSD1 can enhance hepatic lipid deposition (*Kong et al., 2016*; *Masuzaki & Flier, 2004*; *Paterson et al., 2004*). In addition, under non-inducing treatment (Control Diet), Tg animals exhibited high serum glucose accompanied by low insulin and C peptide levels, indicating a consistent Tg insulin secretory defect trend. The related parameters indicated that the mice in the Tg HFHS group were in an early pre-diabetic stage accompanied by insulin resistance (Table 1). The

**Table 1  Serological parameters of mouse models related to pre-diabetes.**

| Parameter | Tg HFHSD | Nc HFHSD | Tg ControlD | Nc ControlD |
|---|---|---|---|---|
| Diabetes phenotype | | | | |
| GLU (mmol/l) | 5.81 ± 1.26 | 2.42 ± 0.41 | 5.40 ± 0.78 | 2.36 ± 1.99 |
| HDL-C (mmol/l) | 1.49 ± 0.25 | 2.12 ± 0.10 | 1.65 ± 0.09* | 2.15 ± 0.04 |
| LDL-C (mmol/l) | 0.41 ± 0.06 | 0.59 ± 0.12 | 0.72 ± 0.24 | 0.50 ± 0.03 |
| Insulin resistance | | | | |
| COR (ng/ml) | 11.21 ± 3.39 | 10.62 ± 2.52 | 12.07 ± 3.05 | 4.75 ± 1.38 |
| TG (mmol/l) | 0.52 ± 0.35 | 0.12 ± 0.03 | 0.17 ± 0.04 | 0.08 ± 0.02 |
| Insulin secretion | | | | |
| INS (uIU/ml) | 19.28 ± 4.92 | 13.50 ± 0.76 | 12.88 ± 1.87* | 18.55 ± 0.34 |
| C-P (mmol/l) | 0.81 ± 0.15 | 1.39 ± 0.69 | 0.71 ± 0.10 | 0.75 ± 0.11 |

**Notes.**

Abbreviations: GLU, glucose; HDL-C, high-density lipoprotein cholesterol; LDL-C, low-density lipoprotein cholesterol; COR, corticosterone; TG, triglyceride; INS, insulin; C-P, C peptide.

Significant differences between the transgenic group (Tg) and the control group (Nc) are indicated by *$p < 0.05$. Tg: $n = 5$; Nc: $n = 3–6$. The data are presented as the mean ± SEM.

mice in the Tg ControlD group were also in the pre-diabetic stage, but their condition was less pronounced than that of the Tg HFHS group. The mice in the Tg ControlD group mouse primarily displayed weak insulin secretion (Table 1).

### Hepatic pathology

Representative livers of the triple-transgenic mice and control mice that received the different diets were photographed (Fig. 5A). Hematoxylin-eosin (HE) staining showing the hepatic adipose deposition in the four treatment groups is presented in Fig. 5A. In total, the areas of 11,000 fat bubbles in non-contiguous sections from 36 mice from the four groups were statistically analyzed (Figs. 5B and 5C). The results indicated that hepatic adipose deposition was more severe in the Tg and HFHS groups (*Ruan et al., 2016*). Specifically, in the two groups of mice that were not fed the HFHS diet (Control diet), the percentage of lipid deposition vacuoles with areas ''<1,500'' was nearly 45% in the Tg animals and only 10% in the Nc animals. There were large numbers of small hepatic adipose vacuoles in Tg animals (Fig. 5B). In the two groups of mice that were fed the HFHS diet, the percentages of lipid deposition vacuoles with areas ''<1,500'', ''1,500–5,000'' and ''5,000–10,000'' were all larger in Tg animals than in Nc animals. Based on this observation, we supposed that for a period of time after HFHS induction, the areas of hepatic adipose vacuoles become enlarged; from this, we deduced that Tg may increase the number of smaller adipose ''points'' (Fig. 5B), whereas HFHS induction can increase the areas of small vacuoles (Fig. 5C).

### Pancreatic pathology

For pathological evaluation of pancreatic tissue, 74 HE-stained islets (magnification: 200×) sections and 55 immunohistochemical sections were prepared from the tail portion of pancreases obtained from a total of 29 mice. Hematoxylin-eosin (HE) staining of tissue from the four groups of animals shows the size of the pancreatic islets (Fig. 6A). In the Tg HFHS group, the areas of the islets are very large compared with those in the Nc

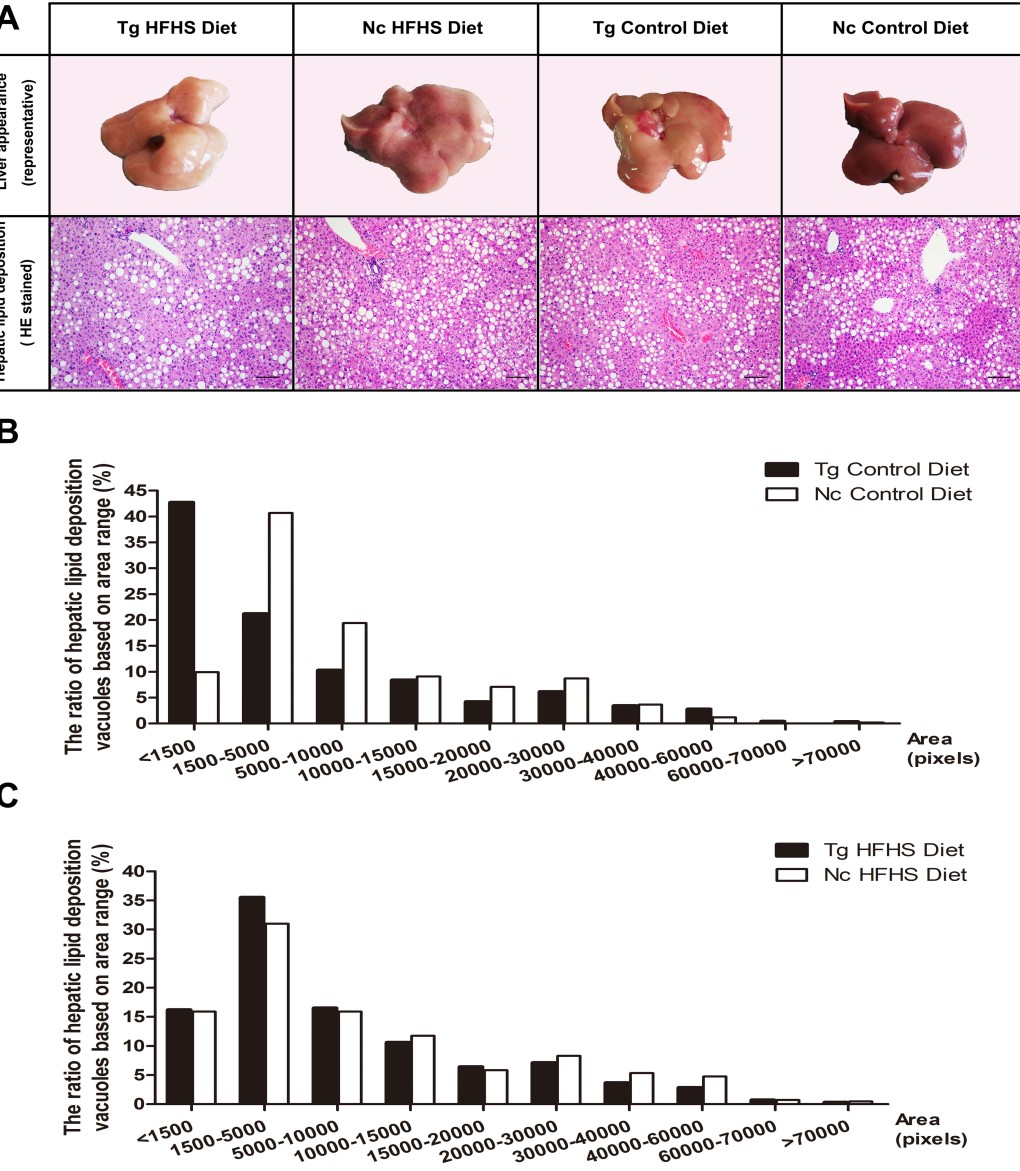

**Figure 5 Liver tissue (hepatology) pathology. (A)** Representative views of the surface of the liver in normal control and triple-transgenic mice and hematoxylin-eosin (HE) staining of the left liver lobe of the four treatment groups is shown. Tg, $n = 5$; Nc, $n = 3–6$. Magnification: $400\times$. Scale bar = 100 μm. The photo of the liver was captured by first author Siyuan Kong. (B) The ratio of hepatic lipid deposition vacuoles based on area range in the two groups of animals that received the Control diet. $n = 3–5$. (C) The ratio of hepatic lipid deposition vacuoles based on area range in the two groups of animals that received the HFHSD diet. $n = 5–6$. The statistical analysis was performed on results obtained using ImagePro Plus v. 6.0. The unit of area was the pixel.

HFHS group (Figs. 6A and 6B). In animals that did not receive the HFHSD, the islet areas appear smaller in the Tg group than in the Nc group (Fig. 6A). The corresponding average pancreatic areas (sum of the number of pixels of each group) were calculated (Fig. 5B). The average area was significantly smaller in the Tg ControlD group than that in the Nc Control

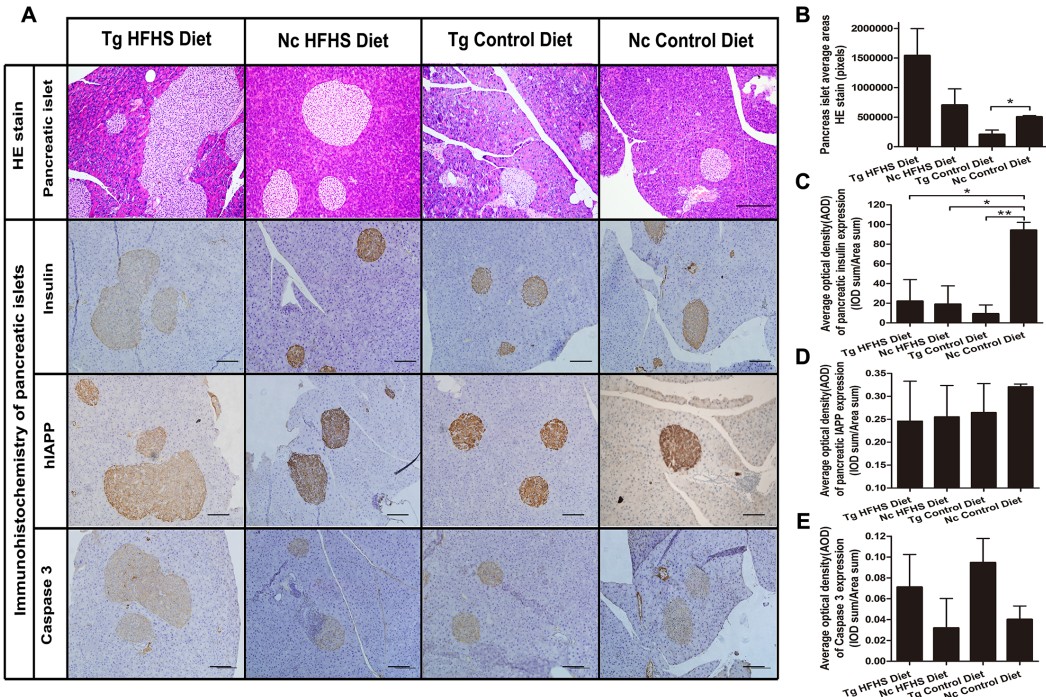

**Figure 6 Pathology of pancreatic islets of Langerhans.** (A) Hematoxylin-eosin (HE) staining of pancre-
atic islets of animals in the four groups. Magnification: 200×. Scale bar = 150 μm. Immunohistochem-
istry of pancreatic islets for insulin, IAPP and caspase 3. All sections were obtained from the pancreatic
tail. Tg: $n = 5$; Nc: $n = 3–6$. Magnification: 200×. Scale bar = 100 μm. (B) The pancreatic islet average
areas (sum pixels of each) from the four groups were calculated. (C–E) The AODs include pancreatic in-
sulin expression density (C); total IAPP expression density (D); and caspase 3 expression density (E). The
significance levels are * $p < 0.05$ or ** $p < 0.01$. P values were calculated using Student's $t$-test.

group (* $p < 0.05$), mostly because the concerted action of hIAPP and CHOP induced
stress and apoptosis and led to islet damage or hypotrophy. In addition, HFHSD strongly
promoted the hyperplasia of pancreatic tissue. Correspondingly, the compensatory effect
was enlarged under the concerted action of hIAPP and CHOP, resulting in an increase in
the areas of islets in the Tg HFHSD group.

The results obtained by immunohistochemistry of pancreatic islets mainly reflected
the islet insulin secretion status (Fig. 6C), islet IAPP deposition density (Fig. 6D) and
islet apoptosis (Fig. 6E). Compared with the Nc ControlD group, the average insulin
secretion status of the other three groups was reduced (Fig. 6C, * $p < 0.05$ or ** $p < 0.01$).
Tg ControlD animals showed reduced insulin secretion and damaged islets (Fig. 6C, Tg
ControlD group<Nc ControlD, ** $p < 0.01$), and HFHSD resulted in enlargement of the
islet area (as a result, Tg HFHSD insulin intensity was low) (Fig. 6B) (*Hu, 2014*; *Ruan et
al., 2016*). The AOD of pancreatic IAPP expression shows the IAPP average accumulation
density of the animals in the four groups (Fig. 6A). The IAPP AOD values were negatively
related to the area (Fig. 6D). Interestingly, the average expression of caspase 3 was higher
in the Tg animals under both diets (Fig. 6E). The concerted action of hIAPP and CHOP
may lead to apoptosis and increased expression of the apoptosis marker caspase 3 in islets.
## DISCUSSION

### The insulin resistance- and insulin secretion-related three-gene model provides a valid basis for a diabetes model that mimics the pathology of diabetes

Several hIAPP-overexpressing single-transgenic rodents of different strains (*Butler et al., 2004*; *Butler et al., 2003*; *Matveyenko & Butler, 2006*) and two types of 11βHSD-1 single-transgenic mice (fat-specific overexpression and liver-specific overexpression) have been reported (*Masuzaki et al., 2001*; *Paterson et al., 2004*). However, the pathogenesis of type 2 diabetes is complex. Most of the previously reported transgenic mice can be used to evaluate the effects of only one factor. It is known that peripheral insulin resistance and impaired insulin secretion are two of the major pathological changes associated with T2DM. Altering insulin resistance and insulin secretion-related gene expression will make the diabetes model more closely mimic the pathology of diabetes. The specific mechanisms of the three genes addressed in our work and the associated model have been discussed in previous reports (*Kong et al., 2016*; *Kong et al., 2015*). As is known, the representative morphological change in pancreatic islets of Langerhans in T2DM is intracellular and extracellular amyloid deposition (*Costes et al., 2013*; *Hull et al., 2013*; *O'Brien et al., 1993*). These deposits consist of human islet amyloid proteins derived from islet amyloid polypeptide (hIAPP) (*Hull et al., 2013*; *O'Brien et al., 1993*). Amylin precipitation overload in islet β-cells can lead to ERS and to the unfolded protein response (*Kayed et al., 2004*; *Meier et al., 2007*). However, although islet amyloid associated with diabetes has been found in humans, monkeys, and cats, it has not been found in rodents (*Johnson et al., 1992*; *Knight, Hebda & Miranker, 2006*). Therefore, we attempted to introduce humanized hIAPP into the rodent transgenic model. When the protein is overexpressed, β-cells become exhausted in response to the deposition of the unfolded protein, leading to apoptosis (*Höppener, Ahrén & Lips, 2000*; *Höppener & Lips, 2006*). Accumulating evidence suggests that islet amyloid deposits may play a significant role in the progressive reduction in the number of insulin-producing cells and in the deterioration of islet function that occurs in diabetes (*Westermark et al., 1987*). In addition, CHOP, which is a transcription factor associated with endoplasmic reticulum stress, is also a direct upstream factor with effects on apoptosis (*Oyadomari & Mori, 2004*) because the CHOP apoptosis pathway in islet β-cells is induced by endoplasmic reticulum stress (*Oyadomari, Araki & Mori, 2002*). CHOP gene knockout diabetic mice showed delayed ER stress and apoptosis (*Oyadomari et al., 2002*). Therefore, we hypothesized that co-expression of hIAPP and CHOP in the pancreas would increase the apoptosis of β-cells (*Höppener, Ahrén & Lips, 2000*; *Johnson et al., 1992*; *Matveyenko & Butler, 2006*; *Oyadomari, Araki & Mori, 2002*; *Oyadomari et al., 2002*; *Oyadomari & Mori, 2004*). Moreover, 11β-HSD1 plays an important role in insulin resistance (*Masuzaki & Flier, 2004*; *Peng et al., 2016*; *Pereira et al., 2012*). Thus, because it brings together insulin resistance and insufficient secretion of insulin, the multi-transgenic mouse model prepared using the tissue-specific polycistronic system described in this work represents an ideal animal model for pre-diabetes mellitus. Because multiple genes are involved in the insulin resistance and the insulin secretion pathways, this

model offers distinct advantages compared to the single transgenic diabetic mouse model (*Lee & Cox, 2011*).

## A combination of multi-transgene expression and HFHSD induction leads to early-stage T2DM with obvious insulin resistance, fatty liver and damaged pancreatic islets

Basing on the comparisons made in this work, the triple-transgenic (Triple-Tg) mice maintained on an HFHS diet were a better model of T2DM than the other transgenic animals that were tested in this study. In young triple-transgenic mice (13 weeks of age), no abnormalities in plasma glucose levels were found. Diet induction was therefore used to mimic the disease induction process of human T2DM. The triple-Tg mice were fed a HFHSD (*Nath, Ghosh & Choudhury, 2016*; *Winzell & Ahrén, 2005*). The results of this study showed that triple-Tg mice fed a high-fat and high-sucrose diet showed early symptoms of diabetes including obesity, impaired glucose tolerance, insulin resistance, abnormal insulin secretion and slightly elevated plasma glucose levels (*Masuzaki et al., 2001*; *Paterson et al., 2004*; *Winzell & Ahrén, 2005*). After several months of induction, the triple-transgenic mice fed a HFHSD had developed early-stage diabetes. The results showed that the IPGTT curve of transgenic mice induced by HFHSD was higher than that of control mice. That is to say, glucose tolerance decreased and insulin resistance occurred earlier in transgenic mice induced by HFHSD than in the other three groups of animals.

Especially at 190 days, the IPGTT AUC of the Tg HFHS group showed a significant change, indicating that insulin resistance was severe in this group. For the control wild-type animals, the trends were increased slightly, but the plasma glucose levels of these animals were still below 15 mmol/l and were less than those of the Tg HFHSD group (more than 15 mmol/l).With respect to the serological parameters of the two groups (HFHS, Control), the glucose level was slightly higher and the C peptide level was lower in triple-transgenic mice than in the corresponding Nc animals. A series of parameters showed a disordered tendency. These signaling pathways are closely related to insulin resistance and β-cell dysfunction (*Xia et al., 2015*; *Yang et al., 2015*). With respect to liver fat deposition, the results for the Tg groups were consistent with previous results obtained in mice with liver overexpression of 11βHSD-1 (*Paterson et al., 2004*), although the phenotypes of both Tg groups were somewhat more severe. To demonstrate liver tissue pathology, we show typical individual livers with progressive disease damage. Preliminary hepatic phenotype analysis showed that transgene expression can accelerate the disease onset process. The Tg effect appeared to increase the number of hepatic adipose vacuoles (Fig. 5B). Moreover, the HFHS diet resulted in an increase in the volume of adipose bubbles (Fig. 5C). Based on the observed phenotypes and the results of HE staining, all of the C57BL/6J mice in the study, regardless of treatment, suffered to a certain degree from hepatic adipose deposition when sacrificed at ∼42 weeks of age. Wild-type C57BL/6 mice tend to develop metabolic syndrome in old age (at 25–78 weeks) (C57BL/6J information from the Jackson laboratory), leading to the production of a measure of fat deposition in the livers of the animals in the negative control group (indicating that these mice are susceptible to type 2 diabetes). In the pancreas, AOD can be used to quantify the average expression density indicated by

immunohistochemistry and thereby to determine the gene expression status of the tissue per unit area (pixel) (*Li et al., 2015*; *Ruan et al., 2016*). In the Tg ControlD group, insulin expression was reduced because the islets may have been destroyed due to the long-term effects of Tg. Although hIAPP may be overexpressed at the early stage, later the AOD of IAPP was very small, likely mostly due to the destruction and enlarged area of islets. The reasons for the reduction in the AOD of insulin and IAPP in the two treatment groups on HFHS diets are likely to be different. The Nc HFHS group was reduced due to islet hyperplasia and increased size. In the Tg HFHSD group, the mechanism may be more complex, merging the role of Tg and HFHS. HFHS can not only enlarge the area of islets, creating hyperplasia, it can also increase insulin secretion, leading to hyperinsulinemia at earlier times as well as to the destruction of islets, slowly reducing insulin or IAPP secretion at later periods when islet overload occurs (*Ruan et al., 2016*; *Winzell & Ahrén, 2005*; *Xia et al., 2015*; *Yang et al., 2015*). However, Tg can also lead to accumulation of IAPP (HFHS can aggravate human IAPP deposition); when the cells are overloaded with unfolded protein, the islets become stressed, leading to damaged islets with reduced insulin and IAPP secretion (*Kayed et al., 2004*). Figures 6C and 6D illustrate the general average results obtaining under different Tg and HFHS conditions that had synergistic or antagonistic effects during the induction period. It is possible that the patterns exhibited by individual mice mimic the complex variations observed in human individuals with diabetes. Many phenotypic and symptomatic differences and some similarities in pathogenesis are also observed in diabetic patients. In triple-Tg mice, caspase 3 expression was higher, so the apoptotic effect was relatively obvious. After induction by HFHSD, although the islet areas were enlarged, these effects were significantly enhanced, leading to severe islet damage and insulin secretion deficiency.

## The advantages and disadvantages of multi-transgenic modeling of polygenic susceptibility diabetes

This study shows that the triple-transgene combined with a HFHSD diet strategy produces mice with increased fat deposition, hepatic lipid deposition, islet dysfunction, hyperinsulinemia and insulin resistance. To develop a model of diabetes that fully recapitulates the end organ manifestations of T2DM with high plasma glucose is the ultimate goal. Here, we achieved the primary goal of early-stage diabetes demonstrating insulin resistance, islet cell survival dysfunction and hyperinsulinemia. We compared our triple transgenic mouse with the widely used leptin-deficient db/db mouse and the ob/ob mouse (*Drel et al., 2006*). Firstly, for the molecular level modeling principle, the ob/ob or db/db mouse are spontaneous models, which are deficient in leptin (ob/ob) or leptin receptors (db/db) (*Drel et al., 2006*). Their abnormalities focus on leptin receptor resistance or desensitization, which just considers a single action pathway. Also, it was not directly reacted with the diabetes genesis signal pathway (*Wang, Chandrasekera & Pippin, 2014*). In addition, genes linked to human T2DM form genome wide association studies (GWAS) of human diabetes do not include genes of leptin and leptin receptor (*Wang, Chandrasekera & Pippin, 2014*). However, we consider this and try to consider the two most important pathogenetic mechanisms linked to human type 2 diabetes, i.e.,

insulin secretion and insulin resistance. Secondly, for the diabetic related symptoms, the ob/ob and db/db mice develop severe hyperglycemia and obesity. They manifest some T2DM-like characteristics. But these two models display only severe early-onset obesity (4 week-old), which is not consistent with the moderate obesity developed in humans later in life (*Drel et al., 2006*; *Wang, Chandrasekera & Pippin, 2014*). In humans, development of T2DM is slow and will be asymptomatic for many years. Although we regard that our triple transgenic mouse was more like pre-diabetes, these models may better recapitulate the characteristics of late-onset moderate obesity and hyperglycemia typical of humans. It was also reported that the ob/ob and db/db mice had high levels of HDL, which was contrary to humans with low HDL (*Wang, Chandrasekera & Pippin, 2014*). In the triple transgenic mouse, we also observed reduced HDL. Thirdly, for the onset of complications, the ob/ob and db/db mice had obvious complications. They had severe hyperglycemia. Long time hyperglycemia was the culprit of organ parafunction (*Drel et al., 2006*). Their complications include cardiovascular disease, renal disease, retinal disease and neuropathy (*Kashyap et al., 2015*). The diabetes incidence of our triple transgenic mouse was early-stage and light. Plasma glucose levels were not very high. There were no obvious complications in the transgenic mouse. Although the transgenic mice did not achieve the strong obvious effects of ob/ob or db/db mice, the multi-transgenic modeling methodology was initially proved to be valuable and could be further optimized in the future for more complicated disease modeling researches. Ideal symptoms of concomitant manifestations may appear when the ultimate model of hyperglycemic diabetes is generated.

The research presented here describes a method for preparing, by genetic modification, an animal model of diabetes that promotes insulin resistance accompanied by islet β cell apoptosis (*Kong et al., 2016*; *Kong et al., 2015*). The advantage of this research is that it uses multiple specific genes to produce a model of diabetes (reverse genetics methodology).

In future work, it would be very interesting to conduct transcriptomic and proteomic analyses of the liver and pancreatic tissue of transgenic mice to determine how transgene expression affects the compensatory proliferation pathway, stress-induced apoptotic pathways and complications (*Li et al., 2015*; *Xia et al., 2015*; *Yang et al., 2015*).

## CONCLUSIONS

An early-stage diabetic mouse model represented by the triple-transgenic 11β-HSD1-CHOP-hIAPP mouse was successfully generated. This research connects susceptible genes, plasma glucose levels, changes in weight, and physiological and biochemical histopathological features. A polygene-modified animal model offers an efficient way to ideally mimic human diseases for which there is susceptibility.

## ACKNOWLEDGEMENTS

The authors thank Ms. Caoqun Wang for primary stage transgenic vectors' construction, Mr. Tao Guo for helping us to analyze the data on transgenic hIAPP-CHOP mice, Prof. Yanfang Wang for help, as well as Leilei Xin, Jihan Xia, Wenjuan Zhu and Cuiping An for their discussion and assistance.

### Funding

This study was supported by the National Natural Science Foundation of China (grant no. 81770832, 31372276), the National Basic Research Program of China (grant no. 2015CB943100), the National Science and Technology Major Project (grant no. 2016ZX08006-001, 2016ZX08010-003), the State Key Laboratory of Animal Nutrition (grant no. 2004DA125184G1602) and the Agricultural Science and Technology Innovation Program (grant no. ASTIP-IAS05 and ASTIP-IAS-TS-4). The funders had no role in study design, data collection and analysis, decision to publish, or preparation of the manuscript.

### Grant Disclosures

The following grant information was disclosed by the authors:
National Natural Science Foundation of China: 81770832, 31372276.
National Basic Research Program of China: 2015CB943100.
National Science and Technology Major Project: 2016ZX08006-001, 2016ZX08010-003.
State Key Laboratory of Animal Nutrition: 2004DA125184G1602.
Agricultural Science and Technology Innovation Program: ASTIP-IAS05, ASTIP-IAS-TS-4.

### Competing Interests

The authors declare there are no competing interests.

### Author Contributions

- Siyuan Kong performed the experiments, analyzed the data, prepared figures and/or tables, authored or reviewed drafts of the paper, approved the final draft.
- Jinxue Ruan, Bingjun Hu and Yuzhu Cheng performed the experiments, analyzed the data, authored or reviewed drafts of the paper, approved the final draft.
- Kaiyi Zhang prepared figures and/or tables, authored or reviewed drafts of the paper, approved the final draft.
- Yubo Zhang conceived and designed the experiments, authored or reviewed drafts of the paper, approved the final draft.
- Shulin Yang conceived and designed the experiments, analyzed the data, contributed reagents/materials/analysis tools, authored or reviewed drafts of the paper, approved the final draft.
- Kui Li conceived and designed the experiments, contributed reagents/materials/analysis tools, authored or reviewed drafts of the paper, approved the final draft.

### Animal Ethics

The following information was supplied relating to ethical approvals (i.e., approving body and any reference numbers):

All procedures were approved by the Animal Care and Use Committee of the Germplasm Resource Center (Institute of Animal Sciences, Chinese Academy of Agricultural Sciences, Beijing, China) (permit no. ACGRCM 2013-035).
## Data Availability

The raw data are provided in the Supplemental Files.

## Supplemental Information

Supplemental information for this article can be found online at http://dx.doi.org/10.7717/peerj.4542#supplemental-information.

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
