# Peer review of "Kill two birds with one stone: making multi-transgenic pre-diabetes mouse models through insulin resistance and pancreatic apoptosis pathogenesis"

_PeerJ, doi:10.7717/peerj.4542_

## Round 0.1 · original submission · Major Revisions

Although both reviewers acknowledge the merit of this paper, both of them indicate the necessity to compare the described multi-transgenic mouse model for diabetes to previous models for diabetes and to indicate whether and how the issue about diabetes-related chronic complications has been addressed.

Reviewer 1 ·

Basic reporting

See below

Experimental design

See below

Validity of the findings

See below

Additional comments

In this paper, the authors construct a mouse model able to replicate, simultaneously, the major pathogenetic mechanisms of T2DM: insulin resistance and beta cell defect. The paper is susceptible for publication, but I have some comments that need to be addressed by the authors:

1. Gene acronims employed in the abstract must be explained
2. Introduction lacks of the main findings originated by previous, and pioneristic works with the most relevant single-gene engineered diabetic mice (generalized and tissue-specific insulin receptor KO, HMGA1 KO, etc). These studies should be cited (at line 106), and briefly commented. On the other hand, from line 115, the text anticipates (in too much details!) what will be shown in the Results. This part should be omitted.
3. Please, fix the following references: “Association” and “Statements”, in the text and in the reference list.
4. Please, split the first chapter of the Results section: one for figure 1 and one for figure 2
5. In panels 2A and 2B, the dual transgenic mouse is lacking. In panels 2C and 2D, the single transgenic mouse is lacking. Please, complete the panels and adopt always the same colours.
6. Lane 250. It is not clear why the authors choose to employ the triple TG mice for next experiments.
7. Discussion section is too long and must be drastically reduced.
8. The manuscript needs an extensive revision of English.

·

Basic reporting

The manuscript is clearly written. References to previous work using a porcine model of metabolic syndrome/diabetes are provided. Raw data supporting the figures are provided.

Experimental design

The experimental design is within the scope of Peer J. There is a clear need to develop a model of diabetes that fully recapitulates the end organ manifestations of type 2 diabetes. The authors compare hypothesize that transgenic animals for CHOP (single transgene), CHOP and hIAPP (dual transgene; both coupled to the porcine insulin promoter), and CHOP-hIAPP and 11-beta-hydroxysteroid dehydrogenase type 1 (linked to the porcine apoE promoter). Their hypothesis is that the triple transgene, by targeting various aspects of islet function/cell survival and insulin resistance, provide a more robust model of metabolic syndrome/diabetes than do the single transgenic animals. The methods, combined with this group's previously published methods on the porcine model, are sufficient to understand how the studies were conducted.

Validity of the findings

The authors do find that the triple transgenic mice have slightly higher blood glucose levels in the IPGTT test, and that administration of a high fat-high sucrose diet led to higher glucose levels than the double transgenic animals. Body weights and adipose tissue accumulation were also higher in the high fat triple transgenic animals. There appears to be a modest increase in hepatic lipid deposition in the triple transgenic animal fed a high fat diet. However, the normal diet control mice also had significant hepatic steatosis. The transgene apparently increases microvesicular steatosis, whereas the high fat diet increases the size of the lipid vacuoles. In general, the high fat diet appeared to increase islet area, whereas the transgene decreased islet size. There do not appear to be statistically significant differences in caspase 3 activity among the 4 groups studied.

Additional comments

The authors clearly show that the triple transgene, combined with high fat diet, produces heavier animals with increased fat deposition, hepatic lipid deposition, and insulin resistance than the single transgenic animals. However, the conclusion that the triple transgenic animal provides a better model of diabetes is premature. The mice were studied for 190 days, which would be a long time to use this model for future studies on pathogenesis or interventions. There are two issues that should be addressed in the discussion--first, there are other models of type 2 diabetes (such as the leptin-deficient db/db mouse or the ob/ob mouse). How does the triple transgenic animal compare with these models? If the triple transgenic animal is a better model than other type 2 diabetes models, this should be discussed. Second, one of the leading problems in diabetes research is the lack of an animal model that recapitulates the end organ manifestations of diabetes, i.e. cardiovascular disease, renal disease, retinal disease, and neuropathy. It was mentioned that the heart and kidney weights of the triple transgenic animals were higher than that of normal mice--were these normalized to body weight? Is there any evidence of cardiovascular or renal disease in this model? This should also be discussed.

---

## Round 0.2 · Minor Revisions

Although the authors have improved their manuscript, there are still several points that need to be improved, and in particular references, and grammar. Please, find the revised file in attachment, with the changes required (sticky notes) and follow suggestions. These points need to be seriously addressed before considering publication.

Reviewer 1 ·

Basic reporting

No comment

Experimental design

No comment

Validity of the findings

No comment

Additional comments

The authors have conceptually improved their manuscript, however there are still some issues that deserve to be well addressed.
1. There is still no mention of the IR knock-out mice, which were the first genetically engineered mouse models for diabetes.
2. English still needs improvement in the newly added parts – these sections should be revised.
3. References need to be double-checked, especially if the authors used an automatic system. Line 51, Association?, line 60, HH? – this is not correct. Line 87, the reference is wrong. Authors should check their cited references according to the context. These are just a couple of examples to take into account.

·

Basic reporting

The revised manuscript has been edited for grammar. Additional references have been added. Data tables are complete.

Experimental design

No comment.

Validity of the findings

No comment

Additional comments

In this revised manuscript, the authors provide several important clarifications. In the revised discussion (lines 575-600) the authors emphasize that the triple transgenic model is employed to study metabolic syndrome and early manifestations of diabetes--insulin resistance, islet cell dysfunction, and hyperinsulinemia. Important comparisons with other commonly used models of diabetes (the ob/ob and db/db mouse) have been added.

---

## Round 0.3 · accepted · Accept

The authors have satisfactorily addressed the issues raised by the Editor.